# Mast Cell Tryptase and Carboxypeptidase A3 in the Formation of Ovarian Endometrioid Cysts

**DOI:** 10.3390/ijms24076498

**Published:** 2023-03-30

**Authors:** Dmitri Atiakshin, Olga Patsap, Andrey Kostin, Lyudmila Mikhalyova, Igor Buchwalow, Markus Tiemann

**Affiliations:** 1Research and Educational Resource Centre for Immunophenotyping, Digital Spatial Profiling and Ultrastructural Analysis Innovative Technologies, Peoples’ Friendship University of Russia, 117198 Moscow, Russia; 2Research Institute of Experimental Biology and Medicine, Burdenko Voronezh State Medical University, 394036 Voronezh, Russia; 3Avtsyn Research Institute of Human Morphology, 117418 Moscow, Russia; 4Institute for Hematopathology, 22547 Hamburg, Germany

**Keywords:** tryptase, carboxypeptidase A3, mast cells, ovarian endometrioma, specific tissue microenvironment

## Abstract

The mechanisms of ovarian endometrioid cyst formation, or cystic ovarian endometriosis, still remain to be elucidated. To address this issue, we analyzed the involvement of mast cell (MC) tryptase and carboxypeptidase A3 (CPA3) in the development of endometriomas. It was found that the formation of endometrioid cysts was accompanied by an increased MC population in the ovarian medulla, as well as by an MC appearance in the cortical substance. The formation of MC subpopulations was associated with endometrioma wall structures. An active, targeted secretion of tryptase and CPA3 to the epithelium of endometrioid cysts, immunocompetent cells, and the cells of the cytogenic ovarian stroma was detected. The identification of specific proteases in the cell nuclei of the ovarian local tissue microenvironment suggests new mechanisms for the regulatory effects of MCs. The cytoplasmic outgrowths of MCs propagate in the structures of the stroma over a considerable distance; they offer new potentials for MC effects on the structures of the ovarian-specific tissue microenvironment under pathological conditions. Our findings indicate the potential roles of MC tryptase and CPA3 in the development of ovarian endometriomas and infer new perspectives on their uses as pharmacological targets in personalized medicine.

## 1. Introduction

Endometriosis is a chronic disorder that is characterized by the growth of endometrial-like tissue outside the uterine cavity, and it is found in approximately 10% of women of reproductive age [1,2,3]. Currently, endometriosis presents an increased number of different variants, and there is a tendency to use organ-preserving surgical interventions as treatment options for this pathology. In addition to chronic pain, dysmenorrhea, and infertility, endometriosis results in conditions with an increased risk of ovarian cancer development [4,5,6]. Despite progress in understanding the biological essence of endometriosis, its pathogenetic mechanisms still remain undiscovered [7,8,9]. The results of ongoing studies have indirectly supported the crucial significance of the stromal and immune landscapes of the local tissue microenvironment for the onset and progression of the disease [10,11,12]. There is evidence that the chronic inflammatory process considerably increases the risk of ovarian endometriosis development [13]. It has recently been demonstrated that the local tissue microenvironment of endometrioid lesions is characterized by features of cytokine and chemokine profiles, which promote a mast cell (MCs) recruitment with a subsequent differentiation [14]. In particular, an increased level of SCF and mRNA CCL2 is formed, which is combined with an increase in the mRNA of the corresponding KITLG and CCR1 receptors, as well as an increase in the mRNA level of VCAM1, CMA1, and carboxypeptidase A3 (CPA3). At the same time, the co-cultivation of MCs with endometriotic epithelial cells and endometrial stromal cells leads to the increased production of pro-inflammatory and chemokinetic cytokines [14], which may explain the pathogenesis of the special properties of the integrative buffer metabolic environment. The special role of MCs is associated with the functional features of their integration into both the immune and stromal landscapes of the tissue microenvironment [15,16,17,18,19]. MCs were first identified by Paul Ehrlich and described in his doctoral dissertation in 1878 [20,21]. Further research has shown the critical translational significance of MC population features in personalized medicine [22]. MCs have a wide receptor range and a rich arsenal of mediators, representing three groups, including pre-formed mediators, de novo synthesized lipid mediators, and cytokines/chemokines [23,24,25,26]. Among these preformed mediators, specific proteases are of a special interest, allowing for the classification of human MCs into several subpopulations [26,27,28]. Tryptase, chymase, and CPA3 are polyfunctional components of the secretome, which can be accumulated in large quantities in the composition of secretory granules [29,30,31,32,33,34]. MCs are key cells that are involved in both innate and adaptive immune processes, which are essential in physiological and pathological conditions, including allergies, tumor, cardiovascular, and autoimmune and respiratory diseases, and in the progression of fibrosis and many others [18,24,35,36,37]. From these perspectives, the role of MCs in the pathogenesis of endometrioid cysts becomes obvious, highlighting the biological effects of specific MC proteases. There have been a number of histochemical, immunohistochemical, and electron microscopy studies on the MCs in human ovarian endometriomas that were performed previously, which found the fundamental potential of MC involvement in the development and progression of a pathology, including in the development of fibrotic changes [38,39,40]. It has been demonstrated that an increased number of chymase- and tryptase-positive MCs in endometriosis foci can contribute to the development of pain and hyperalgesia, due to a direct effect on nerve fibers [41]. However, the studies on MCs that were performed previously had limitations, since they included only description of the number and degree of the MC degranulation, and since the analysis of the CPA3 expression was not performed earlier. The present study deals with a histo- and cytographic analysis of MC tryptase and CPA3 in the formation of endometrioid ovarian cysts.

## 2. Results

### 2.1. Ovarian Mast Cell Population in Patients without Pathological Changes

In the ovaries without pathological changes, the MCs were located mainly in the medulla, represented by the loose fibrous connective tissue with a large number of vessels, having a wide lumen (Figure 1, Table 1). Isolated MCs were detected under the ovarian surface epithelium in the albuginea (Figure 1b). In the medulla, the MCs were localized near the elements of the vascular bed, were rather large in size, and were located in close contact with the cells of the cytogenic stroma (Figure 1). Accompanying the vessels of the microvasculature, the MCs provided the migration of other immunocompetent cells, including lymphocytes and granulocytes, into the ovarian stroma (Figure 1c–e). In numerous MCs, secretory granules were located along the cell periphery (Figure 1j,k). MCs were found in corpus albicans, often localized at the border with the cytogenic stroma (Figure 1f). The MCs in the control group most commonly contained both specific proteases, tryptase and CPA3 (Figure 1v). Both the tryptase and CPA3 were localized preferentially to the secretory granules, which could be up to 1 µm in size (Figure 1). However, certain amounts of specific proteases were also detected in the MC cytoplasm, reflecting the ongoing stages of their biogenesis, as well as the gradual maturation of the secretory granules and the exchange of secretome components, including tryptase and CPA3 (Figure 1j,n,v). Secretory activity was manifested in the excretion of individual granules into the extracellular matrix of the ovarian stroma (Figure 1v). The mechanisms of secretion that were identified by light optics were the secretion of individual secretory granules, transgranulation, exocytosis, and the formation of the loose-lying, nuclear-free fragments of the cytoplasm (Figure 1). Histotopographically, the MCs were predominantly evenly distributed in the stroma, but could migrate to particular areas, apparently for the appropriate regulatory effects. In the stroma, the MCs produced a selective effect through the proteases on nearby cells and within the paracrine zone of influence. There was sometimes the impression of the direct contact of the proteases with the elements of the nuclear membrane of the ovarian stromal cells (Figure 1c–e,g,i,t). The interaction of the MCs with fibroblasts is worth noting, since there were often signs of their contact or location in the paracrine zone, in order to propagate the effects of the tryptase and CPA3. Moreover, the active secretory activity implied a direct effect of the specific MC proteases on the fibroblasts, providing regulatory effects that are related to profibrogenic changes. In some fibroblast-like cells, we recorded the content of tryptase in their nuclei (Figure 1i). Sometimes an MC was contacted with several fibroblasts at once, regulating the state of each of them (Figure 1g–m). The secretion of tryptase, as well as of CPA3, was also targeted to the nuclei of the other cells of the cytogenic stroma (Figure 1n–q,s–u). The MCs were actively secretory, as evidenced by the completion of the life cycle in some of them, with a complete loss of proteases (Figure 1r). Moreover, it was obvious that these postcellular formations did not contain a nucleus, which resulted in a gradual depletion of the secretory resource. In a few MCs, their secretory granules were located predominantly in the peripheral layer of the cytoplasm, ensuring the excretion of tryptase into the extracellular matrix using the transgranulation mechanism. In the ovarian MCs, it was possible to observe the successive stages of the mechanism of MC denucleation by gradually shifting the nucleus to the periphery (Figure 1m–r). In addition, pigment cells were quite limited; this apparently was due to an absence of hemorrhages, which resulted from the development of the pathology, including the endometrioid cysts formations.

### 2.2. Ovarian Mast Cells in Patients with Endometrioid Cysts

The development of ovarian endometrioma was accompanied by a significantly increased number of MCs in the organ (Figure 2a and Figure 3a–c, Table 1). According to the calculations, the number of tryptase-positive MCs could be up to 123 per mm^2^ in the examined patients, which is more than 5–10 times higher than the similar parameters in the patients with pathology-free ovaries. In some patients, there were specific areas of the ovary with a high content of degranulating MCs, which indicated that there were active processes for implementing the biological effects of the specific proteases in the loci of the ovarian stroma (Figure 3b). There were practically no MCs in the areas with completed fibrotic changes; the MCs were concentrated in the border zone with the unchanged ovarian stroma tissue (Figure 3b). The largest number of mast cells was observed in the stroma of ovarian endometrioid cysts (OEC) (Table 1).

### 2.3. Histotopographic Features

An increased total amount of MCs in the stroma was accompanied by their migration into the ovarian cortex. In addition, MCs with signs of the secretory activity of specific proteases started to be detected in the perifollicular zone (Figure 2c,d). In some cases, the MCs were in direct contact with the follicular epithelium (Figure 2c,d). The amount of MCs increased in the area of the vessels of the microvasculature, where they were adjacent to the endothelial cells or pericytes (Figure 2g,h,o–q). Cases of MC contact with each other were more frequently detected, having specific physiological effects (Figure 2b,n,r and Figure 3q). We observed diverse variants of this MC interaction, including the phenotypes Tr^+^CPA3^+^ and Tr^−^CPA3^+^ (Figure 4a), Tr^+^CPA3^−^ (Figure 4b), Tr^+^CPA3^+^, and Tr^+^CPA3^−^ (Figure 4c). The active targeted secretory activity of specific MC proteases in relation to certain immunocompetent cells and the cells of the stromal landscape, including fibroblasts, became obvious (Figure 2i,j and Figure 3d–m). The MC communication with other immunocompetent cells, in particular with plasma cells (Figure 2l,m) and eosinophils (Figure 2n), was revealed. The contact of these cells with each other allows for the evaluation of the features of the immune reaction that develops in the limited tissue loci of the ovaries. The MCs could, most commonly, form several contacts with the surrounding cells of the cytogenic stroma (Figure 2k and Figure 3d,e,j,k,m–o). Concurrently, epifluorescence and confocal microscopy detected the formation of MC outgrowths at considerable distances, which was of great significance (Figure 4g–i). It appears that, in this way, the MCs could not only unite the cytogenic stroma cells into a single functional unit, but they could also create large morphogenetic fields to develop the features of a specific tissue microenvironment. In some cases, we observed the spread of secretory granules to large areas of the ovary (Figure 3b). These secretory granules could be selectively accumulated around the vessels of the microvasculature (Figure 2q), or could intensively infiltrate large areas of the tissue microenvironment of the ovary (Figure 3b,b’,c). Apparently, these events are associated with the formation of fibrous tissue and other morphogenetic changes that are involved in the pathogenesis of endometriomas.

### 2.4. Cytotopographic Features

Compared with the norm, MCs of various phenotypes, with specific protease content, were detected within ovarian endometriosis. It was possible to identify CPA3^+^Tr^−^ TK and Tr^+^CPA3^−^ (Figure 4a–d). However, it should be noted that the MCs that migrated to the epithelial lining of the cysts predominantly expressed only tryptase (Figure 4e,f). The content of the granules in the MC cytoplasm also varied. In particular, we could observe tryptase-positive granules, CPA3-positive granules, and granules containing both proteases (Figure 5). The fact that MC overgrowths were formed several micrometers towards neighboring cells was worth attention. It can be assumed that this is a new way of developing the regulatory effects of MCs on the cells of the local tissue microenvironment (Figure 4g–j). The presence of specific MC proteases in such processes supports active tryptase and CPA3 participation in the regulation of the physiological activity of neighboring cells.

Of crucial importance is the detection of tryptase and CPA3 in the nuclei of both cytogenic stroma cells and MCs (Figure 2h, Figure 5d–f and Figure 6k). In the control samples, such phenomena were limited, although this may result from the small number of MCs that were exposed to analysis. The amount of tryptase in the nuclei under the pathology could be significant, as this revealed the diffuse immunopositivity of the nuclear structures (Figure 2h).

The MCs actively migrated into the walls of the endometrioid cysts, reaching the lining columnar epithelium (Figure 4e,f). Moreover, the MC localization in specific areas of the wall was also combined with perivascular distribution and active secretion. Namely, tryptase-targeted secretory activity could be detected in relation to both the epithelium and stroma. Contacts with immunocompetent cells were also manifested (Figure 2k–p). Sometimes, the amount of the secreted tryptase was so high that quite extensive tryptase-positive fields formed around the cells (Figure 2h). MC localization near the vessels initiated the migration of other leukocytes into the specific tissue microenvironment of the ovary (Figure 2o,p). The nucleus position in the MCs was diverse. In some cases, the nucleus was well-detected in the MCs, occupying a central position (Figure 5a,c,d–f). However, in some cells, the nucleus was located on the periphery, and there was the impression that the nucleus exited from the cytoplasm (Figure 5b,g and Figure 6j). In addition, it is necessary to pay attention to the detection of the non-nuclear fragments of MCs: the absence of a nucleus presented throughout the entire thickness of the section; the thickness could reach 7 μm along the Z axis, according to the automatic scanning data (Figure 5h and Figure 6l).

The activity of the specific protease secretion into the extracellular matrix significantly increased during the development of endometriomas. There was an active entry of various MC phenotypes with specific proteases, including the Tr^+^CPA3^−^, Tr^−^CPA3^+^, and Tr^+^CPA3^+^ variants, into the extracellular matrix (Figure 6a,b). Under the localization of the MC granules in the intercellular substance, we revealed their potential for selective specific protease secretion, independent of each other (Figure 6a). Interestingly, MC secretory granules could be located near the karyolemma of the stromal cell nuclei. In some cases, we observed a changed shape of these MC secretory granules, which increased in length towards the secretion locus (Figure 6c). Of special attention is the fact that the MCs manifested cytoplasmic outgrowths of various lengths and thicknesses that were immunopositive to specific proteases, which were directed to particular structures of the tissue microenvironment (Figure 6d–g). There were also MCs with protease secretory pathways that performed through transgranulation mechanisms (Figure 6h,i).

## 3. Discussion

In our study, we have demonstrated the presence of MCs in various areas of the ovary in patients without pathological changes and with endometriomas; these findings support the results of a few previously performed studies, including those with tryptase and chymase immunodetection [42]. The persistent presence of MCs in the ovary results in a specific organ histotopography, since the majority of the MC population resides in the medulla. The development of ovarian endometriomas is accompanied by a significantly increased number of MCs in the organ. It is known that the induction of the pathological focus development in endometriosis is manifested by a change in structural elements, the formation of a pro-inflammatory homeostasis of the local tissue microenvironment, a resistance to apoptosis, and an increased angiogenesis [43]. These phenomena directly or indirectly result from the pro-inflammatory biological effects of tryptase. Namely, the suppression of the activated MC and tryptase secretory activity produced anti-inflammatory and analgesic effects in experimental models that were simulating pain and chronic inflammation, as well as in the clinical treatment of chronic pain [44,45,46].

In the pathogenesis of ovarian endometrioma, it is necessary to consider the active MC migration from extra-organic sources. In this context, the specific cytokine and chemokine profiles of the local tissue microenvironment, which provide the recruitment of MCs from the microvasculature, result in the active entry of MCs to the ovary during the formation of endometrioid cysts [14]. It is known that MCs can be the first among immunocompetent cells to migrate to the area of malignancy [47,48]. MC secretory activity possesses a wide range of biologically active substances that are produced by mediators, and can manifest considerable inducing effects on molecular, cellular, and tissue targets. Our study demonstrates a significant activation of the tryptase and CPA3 secretion by MCs into the local microenvironment of the ovarian stroma. It can be reported that the specific tissue microenvironment leads to the activation of MC degranulation. Different types of stimuli can cause the differential release of MC mediators. Among the preformed mediators, tryptase is the most abundant secretory granule protein in human MCs, and its amount can reach 25% of the total protein content within the cell [49]. Tryptase has a wide range of biological activities and, as demonstrated, regulates immunogenesis, being a link within the innate immunity, provides the inactivation of toxins, and regulates the state of the intraorgan stroma components through the extracellular matrix remodeling; this also includes stimulating the formation of fibrous structures and the progression of fibrosis in preclinical models on animals [33].

The results of recent studies that have investigated the fundamental significance of tryptase entry into the nuclei of other cells [50] are supported by our findings, evidencing the potential, significant morphogenetic effects of MCs on the ovarian stroma. In particular, nuclear tryptase can process the core histones in the N-terminal tail, and thereby regulate transcription processes [51,52]. The provided principle of regulating the epigenetic modification of core histones, which ultimately leads to the restriction of cell proliferation, represents a strategically novel understanding of the biological function of human MC tryptase [52]. As demonstrated, DNA molecules stabilize the enzymatic activity of tryptase; therefore, protease, even in the absence of heparin and the other polyanions of the secretory granules, can regulate the selected events directly in the cell nuclei, for quite a long time [51,52,53]. The close colocalization of CPA3 with tryptase, which can be revealed by its location in the secretory granules, predetermines the involvement of CPA3 in a number of epigenetic effects. In our studies, we first managed to visualize CPA3 in the nuclei structures of other cells.

Data on tryptase’s ability to regulate cell proliferation are of particular interest, since ovarian endometriosis is a hormone-dependent process that is capable of malignancy, with the formation of a true neoplasm. Estrogen is an important mediator of ovarian MC activation [54]. As is known, estrogen can indirectly promote the growth of endometrial lesions through the TNF-α and NGF secretion of MCs [55]. In addition, MC tryptase may be an essential trigger for the secretion of inflammatory cytokines by other cells in the tissue microenvironment [56].

The tryptase effects can be differentiated into pro- or anti-inflammatory at the level of the tissue microenvironment [49,57,58,59,60]. Most commonly, tryptase initiates the development of inflammation, causing an increased permeability of the capillary wall, with an increased intensity of neutrophils’, eosinophils’, basophils’, and monocytes’ migration outside the microvasculature; this is also supported by the results of our research study [61].

In the context of the endometrioid cyst pathogenesis, data on the close tryptase involvement in angiogenesis processes are of great interest [24,62,63]. Tryptase is a part of several mechanisms of new blood vessel growth and differentiation, including inflammation and tumorigenesis [64,65,66]. Concurrently, the formation of new vessels is combined with a pronounced remodeling of the connective tissue and the degradation of the amorphous and fibrous components of the extracellular matrix, due to the tryptase-induced activation of matrix metalloproteinases (MMPs), in particular, MMP-1, MMP-2, MMP-3, MMP-9, and MMP-13, etc. [67,68,69]. The effects of tryptase, in relation to the cells of fibroblastic differon, are manifested by the activation of their migration, mitotic division, the stimulation of the synthesis of collagen proteins, and the predetermining of fibrotic aftereffects [67,70,71].

Tryptase has a high affinity for PAR-2 receptors, potentiating the development of inflammation [72]. The localization of these receptors on the various cells of a specific tissue microenvironment can induce leukocyte migration and edema, etc. In addition, the tryptase-activating effect on the PAR-2 receptors of afferent neurons can produce neurogenic inflammation and the development of pain syndrome. A persistent, increased expression of the PAR-2 receptors in various cells, including, possibly, the ovarian cytogenic stroma, is an essential regulatory mechanism of tryptase in the potentiation of inflammation. Tryptase is able to activate the secretion of pro-inflammatory mediators into the extracellular matrix via the cells of a specific tissue microenvironment, producing an increased background content of a number of cytokines and chemokines [72,73,74]. Tryptase results in the stimulation of histamine liberalization from intracellular depots, which, in turn, causes a new increase in the tryptase secretion. This contributes to the involvement of new MCs in the process of degranulation, providing a realization of the biological effects of histamine over a larger area [75].

In the context of tryptase and carboxypeptidase A3 colocalization under secretion, one can assume both purely specific biological effects and unidirectional ones. The formation of fibrosis has been demonstrated in numerous studies on endometriosis [76,77,78]. Notably, the role of MCs is clearly underestimated when compared to myofibroblasts, macrophages, platelets, and the other participants that are involved in the formation of the profibrotic phenotype of a specific tissue microenvironment [79,80]. The results of our study, as well as those from previous investigations, allow us to consider MCs as important inducing elements for fibrillogenesis and the finalization of the processes of the increasing collagen fibers in ovarian fibrous tissue. This is evidenced by the histotopographic features of MC distribution, reflecting the known direct and indirect effects of MCs on the processes of collagen fibrillogenesis [71]. Previous studies have supported MC active participation in the mechanisms of fibrillogenesis, which is manifested by an inductive effect on the formation of the fibrous component of the tissue microenvironment, primarily in the loci of the local microenvironment of the fibroblastic differon [71].

One can hypothesize a crucial role of CPA3 in the biogenesis of the fibrous component of the extracellular matrix and in extracellular matrix remodeling. On the one hand, a CPA3 complex with chymase can induce the increased mitotic activity of fibroblasts, along with their biosynthetic potential [81]. On the other hand, a CPA3 macro-compound with chymase or proteases, separately, can participate in the modification of procollagen molecules, inducing collagen fibril formation [27,82].

Indirect evidence of the CPA3 involvement in angiogenesis was obtained in the experiment involving *Cpa3^Cre/+^* mice with a deficiency of MC carboxypeptidase. Such a pathology, having various manifestations, caused a decrease in the number of the intraorganic MC population and a delayed revascularization under bone tissue damage simulation [83].

The spatial phenotyping of MCs’ cytological features under pathological conditions revealed the formation of cytological outgrowths that were detected at considerable distances. These structures had a certain similarity with “tunneling nanotubes”, which contribute to the interconnection of MCs with each other and with other cells [84]. We showed that these structures were immunopositive for tryptase and CPA3. From this standpoint, it can be considered that there is a particular similarity of the detected phenotype with the neuronal organization. Experiments to determine the neuronal markers in MCs have led to the discovery of overexpressed genes, when compared to the other representatives of myeloid cells [85]. In particular, these MC genes include MLPH and RAB27B (traffic/lysosomes), MAOB, DRD2, SLC6A3, and SLC18A2 (dopamine system), CALB2 (calcium-related processes), L1CAM and NTM (adhesion molecules), and LMO4, PBX1, MEIS2, and EHMT2 (transcription factors and modulators of transcriptional activity) [85]. In this regard, we can assume that, within endometrioid ovarian cyst formation, MCs are able to acquire, in a determinate sense, a neuro-like morphological phenotype in order to activate the integrating/regulating functions that are required in a particular locus of the local tissue microenvironment.

It should be noted that, when studying the role of mast cells in the ovary, the use of routine histochemical staining, for example, with toluidine blue, is not effective enough, due to the low content of glycosaminoglycans in the secretory material for the development of the metachromasia effect. Therefore, for the identification of mast cells during a pathology examination of tissue specimens, it is necessary to use characteristic markers that are found in most mast cells in their organ-specific population, including specific proteases or CD117. This conclusion is also objective for the pathologist to evaluate the MCs in other organs, in particular, in the gastrointestinal tract [86].

## 4. Materials and Methods

### 4.1. Case Selection

This study was conducted on an OEC biomaterial that was obtained after a cystectomy operation and on endometriosis-associated ovarian tumor biomaterials from 27 patients that were aged from 26 to 49 years (mean 34.86 ± 1.37 years), in the period from 2016 to 2019, in the City Clinical Hospital S.S. Udina, Moscow and the City Clinical Hospital #31, Moscow. A total of three pathologists reviewed the samples independently. Only the cases in which there was a unanimous agreement on the histological diagnosis of OEC were included in this study. In the control group, the biomaterial was obtained from 6 patients of childbearing age who died from diseases that were not associated with endometriosis or malignant neoplasms. This study was conducted in accordance with the principles of the World Medical Association Declaration of Helsinki, “Ethical Principles for Medical Research Involving Human Subjects”, and approved by the Institutional Review Board of the City Clinical Hospital #31, Moscow (approval protocol No. 3, 12 June 2019). The samples were retrieved from the files of the City Clinical Hospital S.S. Udina (Moscow, Russia) and the City Clinical Hospital #31 (Moscow, Russia). Tissue samples were taken for diagnostic purposes. Informed consent was obtained from all the subjects. The samples were qualified as redundant clinical specimens that had been de-identified and unlinked from patient information.

### 4.2. Tissue Probe Staining

The tissue probes that were left over during the routine diagnostic procedure were fixed in buffered 4% formaldehyde and routinely embedded in paraffin. The paraffin tissue sections (2 µm thick) were deparaffinized with xylene and rehydrated with graded ethanols, according to a standard procedure [87].

### 4.3. Immunohistochemistry

For the immunohistochemical assay, we subjected the deparaffinized sections to an antigen retrieval by heating the sections in a steamer, with R-UNIVERSAL Epitope Recovery Buffer (Aptum Biologics Ltd., Southampton, SO16 8AD, UK), at 95 °C for 30 min. The blocking of the endogenous Fc receptors, prior to the incubation with primary antibodies, was omitted, according to our earlier recommendations [88]. After the antigen retrieval and, when required, the endogenous peroxidase was quenched and the sections were immunoreacted with primary antibodies. The list of primary antibodies that were used in this study is presented in Table 1. An immunohistochemical visualization of the bound primary antibodies was performed either with Roche Ventana BenchMark Ultra (Roche Diagnostics, Indianapolis, IN, USA) or manually, according to the standard protocol [87,89]. For the manually performed immunostaining, the primary antibodies were incubated overnight at +4 °C, in optimal dilution (Table 2).

The bound primary antibodies were visualized using secondary antibodies (purchased from Dianova, Hamburg, Germany, and Molecular Probes, Darmstadt, Germany) that were conjugated with Alexa Fluor-488 or Cy3. The final concentration of the secondary antibodies was between 5 and 10 µg/mL PBS. Single and multiple immunofluorescence labeling was performed according to standard protocols [87]. The list of secondary antibodies and other reagents that were used in this study is presented in Table 3.

### 4.4. Controls

The control incubations were: an omission of the primary antibodies or a substitution of the primary antibodies by the same IgG species (Dianova, Hamburg, Germany), at the same final concentration as the primary antibodies. The exclusion of either the primary or the secondary antibody from the immunohistochemical reaction, and the substitution of the primary antibodies with the corresponding IgG at the same final concentration, resulted in a lack of immunostaining. The specific and selective staining of different cells with the use of primary antibodies from the same species on the same preparation is, by itself, a sufficient control for the immunostaining specificity.

### 4.5. Image Acquisition

The stained tissue sections were observed on a ZEISS Axio Imager.Z2 that was equipped with a Zeiss alpha Plan-Apochromat objective 100×/1.46 Oil DIC M27, a Zeiss Objective Plan-Apochromat 150×/1.35 Glyc DIC Corr M27, and a ZEISS Axiocam 712 color digital microscope camera. The captured images were processed with the software programs, “Zen 3.0 Light Microscopy Software Package”, “ZEN Module Bundle Intellesis & Analysis for Light Microscopy”, “ZEN Module Z Stack Hardware” (Carl Zeiss Vision, Aalen, Germany), and submitted with the final revision of the manuscript at 300 DPI. Photomicrographs were obtained in some cases with a confocal microscope, Nikon D-Eclipse C1 Si based on Nikon “Eclipse 90i”. The number of mast cells in the different areas of the ovary was determined per mm^2^ of tissue, using open-source software for the digital pathology image analysis, QuPath [90] (Bankhead P., et al., 2017) (Table 1).

### 4.6. Statistical Analysis

The statistical analysis was performed using the SPSS software package (Version 13.0). The results are presented as means (M) ± m (standard error of the mean). To assess the significance of the differences between the two groups, a Mann–Whitney U test was used.

### 4.7. Data Availability

The authors declare that all the data supporting the findings of this work are available within the article, or from the corresponding authors upon reasonable request.

## 5. Conclusions

MC tryptase and CPA3 play an important part in the development of ovarian endometriomas. The MC functional potential is realized in ovarian pathology due to the specific nature of MCs’ anatomical structure, the organization of a specific tissue microenvironment, and the vascularization parameters that produce favorable conditions, in order to form selective effects on cellular and molecular targets, using secretome components. The systematization of existing morphological knowledge, with an emphasis on the profile of specific MC proteases, can throw a great deal of light on new, individual patterns of the disease development and the nature of the disease progression. The revealed range of the regulatory mechanisms of MCs, through their paracrine and epigenetic effects, highlight a potential of their neuron-like adaptation, providing that further research is performed in this area, with more detailed information needing to be obtained and perspectives to be used in translational medicine. We hope that knowledge on the features of MCs will help to objectify the assessment of the morphological and functional forms of endometriosis, in order to select the optimal treatment options and improve the effectiveness of its conservative therapy.

## Figures and Tables

**Figure 1 ijms-24-06498-f001:**
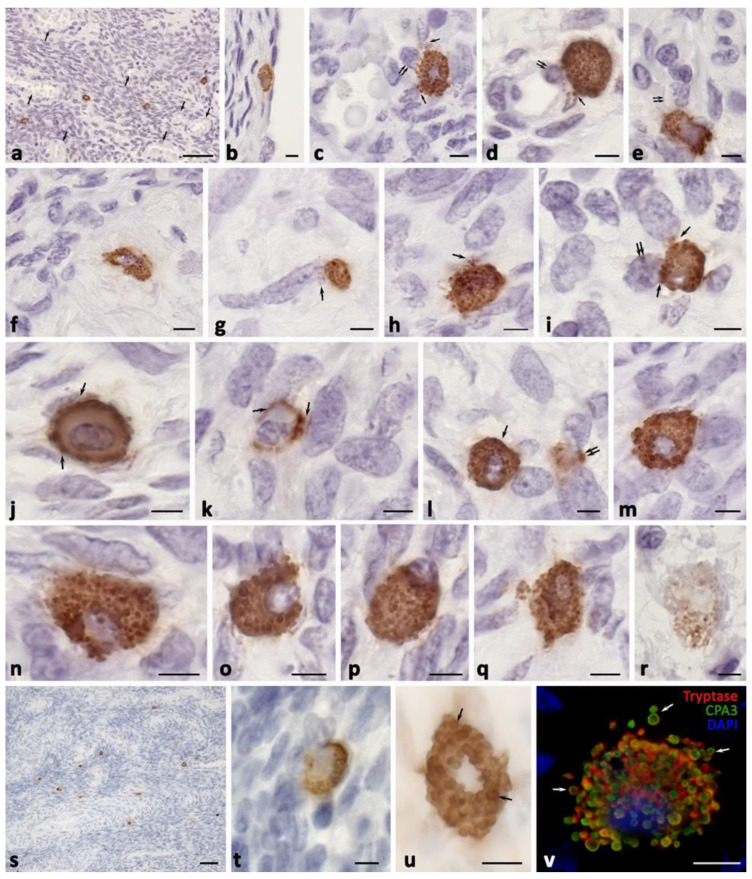
Mast cells of the ovary without pathological changes. (**a**–**r**) Immunohistochemical staining of tryptase, (**s**–**u**) immunohistochemical staining of CPA3, and (**v**) double immunolabeling of tryptase and CPA3. Notes: (**a**) General view of the ovarian medulla. Vessels are well-marked (indicated by an arrow) and there are isolated MCs in the cytogenic stroma. (**b**) MCs on the surface of the protein membrane (the surface epithelium at the site of MC attachment was not preserved). (**c**–**e**) MCs in the perivascular space with targeted tryptase secretion to the endothelium (indicated by an arrow), with induction of transendothelial migration of a lymphocyte (**c**,**d**) (indicated by a double arrow) and neutrophilic granulocytes (**e**) (indicated by a double arrow). (**f**) MCs in the structures of the albuginea. (**g**–**i**) Variants of fibroblast-oriented tryptase secretion in the cytogenic stroma (indicated by an arrow), localization of tryptase in the nucleus ((**i**), indicated by a double arrow). (**j**,**k**) Peripheral arrangement of granules in the cytoplasm, targeted tryptase secretion through the mechanism of transgranulation to cells of the cytogenic stroma (**j**) and fibroblasts (**k**) (indicated by an arrow). (**l**) Tryptase secretion from MCs (indicated by an arrow) and secretory granules (indicated with a double arrow). (**m**–**r**) Sequence of the MC denucleation mechanism by gradual displacement of the nucleus to the periphery (**m**–**p**), with the formation of a nuclear-free cytoplast (**q**) and gradual loss of tryptase content in secretory granules (**r**). (**s**) Location of CPA3-positive MCs in the perivascular cytogenic stroma of the ovary. (**t**) Attachment of CPA3-positive MCs to several stromal cells simultaneously. (**u**) Predominant CPA3 localization in the secretory granules of MCs (indicated by an arrow). (**v**) Colocalization of tryptase and CPA3 in secretory granules, which persists after secretion into the extracellular matrix (indicated by an arrow). Scale: (**a**,**s**) 50 µm, the rest—5 µm.

**Figure 2 ijms-24-06498-f002:**
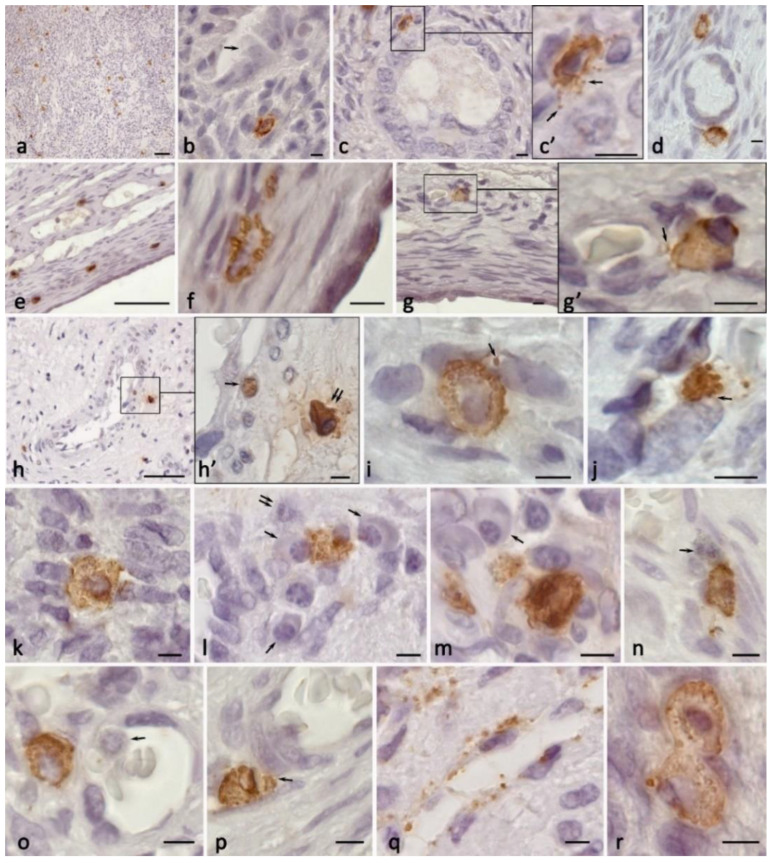
Tryptase-positive MCs in the specific tissue microenvironment of the ovary under the formation of endometrioid cysts. (**a**) An increased MC population in the ovarian medulla. (**b**) Localization of MCs in the tissue microenvironment of the endometrioid cyst (indicated by the arrow). (**c**) Target tryptase secretion to the basement membrane of the follicular epithelium of the secondary follicle (indicated by an arrow). (**d**) MCs in the tissue microenvironment of the primary follicle. (**e**–**g**) Increased number of MCs in the albuginea with active tryptase secretion, including that to the basement membrane of the endothelium ((**g**), indicated by an arrow). (**h**) Perivascular localization of MCs with active tryptase secretion (indicated by a double arrow) with penetration into the nuclei of stromal cells (indicated by an arrow). (**c′**), (**g′**) and (**h′**) are enlarged fragments of (**c**), (**g**) and (**h**), respectively. (**i**,**j**) Target secretion of tryptase to ovarian stroma fibroblasts (indicated by an arrow). (**k**) Simultaneous contacting of MCs with several cells of the cytogenic stroma. (**l**–**n**) Interaction of MC with immunocompetent cells, including plasma cells (indicated by an arrow) and granulocyte (indicated by a double arrow) (**l**,**m**), eosinophil (indicated by an arrow) (**n**). (**o**) Tryptase-induced transendothelial migration of a granulocyte (indicated by an arrow) into the cytogenic stroma of the ovarian medulla. (**p**) Selective effect of tryptase on pericyte (indicated by an arrow). (**q**) Localization of tryptase-positive MC granules in the periendothelial layer of the stroma of the ovarian medulla. (**r**) Contacting MCs. Scale: (**a**,**e**,**h**) 50 µm, the rest—5 µm.

**Figure 3 ijms-24-06498-f003:**
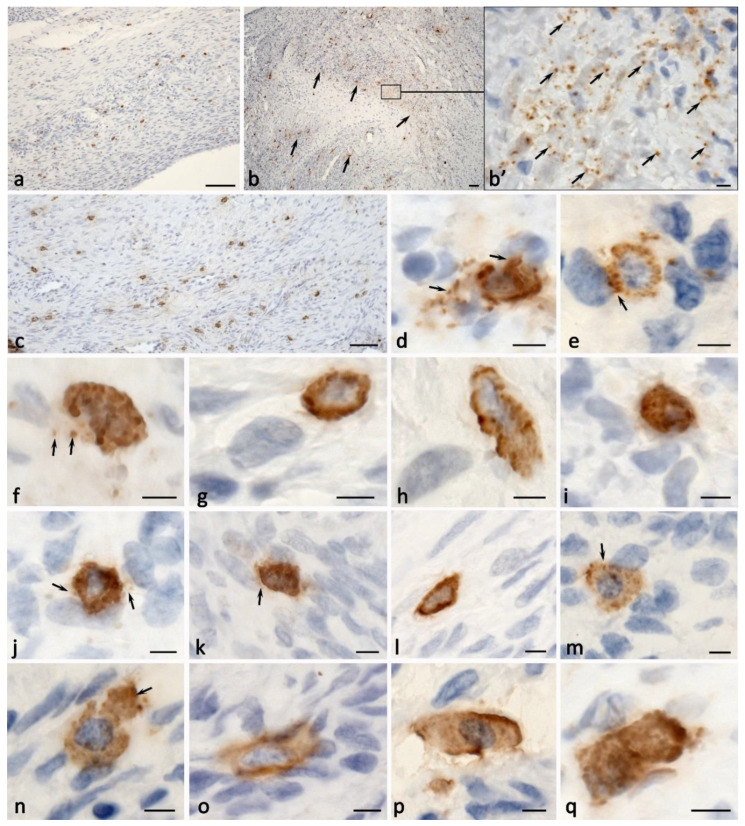
Expression of CPA3 in the population of ovarian mast cells under intraorgan endometrioma development. (**a**) High content of MCs in the wall of the endometrioid cyst. (**b**) Predominant location of MCs and secretory granules bordering with the area of fibrous changes in the ovarian stroma (indicated by an arrow). (**b′**) Enlarged (**b**) fragment, there is a high content of secretory granules in the extracellular matrix (indicated by an arrow). (**c**) Large amounts of MCs in the perivascular connective tissue of the ovarian stroma. (**d**–**m**) Variants of MC colocalization with ovarian stroma fibroblasts, with signs of active targeted secretory activity (indicated by an arrow). (**n**) Release of an autonomous fragment of a protease-positive cytoplasmic region (indicated by an arrow). (**o**) MC contacting with numerous cells of the cytogenic stroma. (**p**) The extracellular matrix remodeling. (**q**) Contacting MCs. Scale: (**a**–**c**) 50 µm, the rest—5 µm.

**Figure 4 ijms-24-06498-f004:**
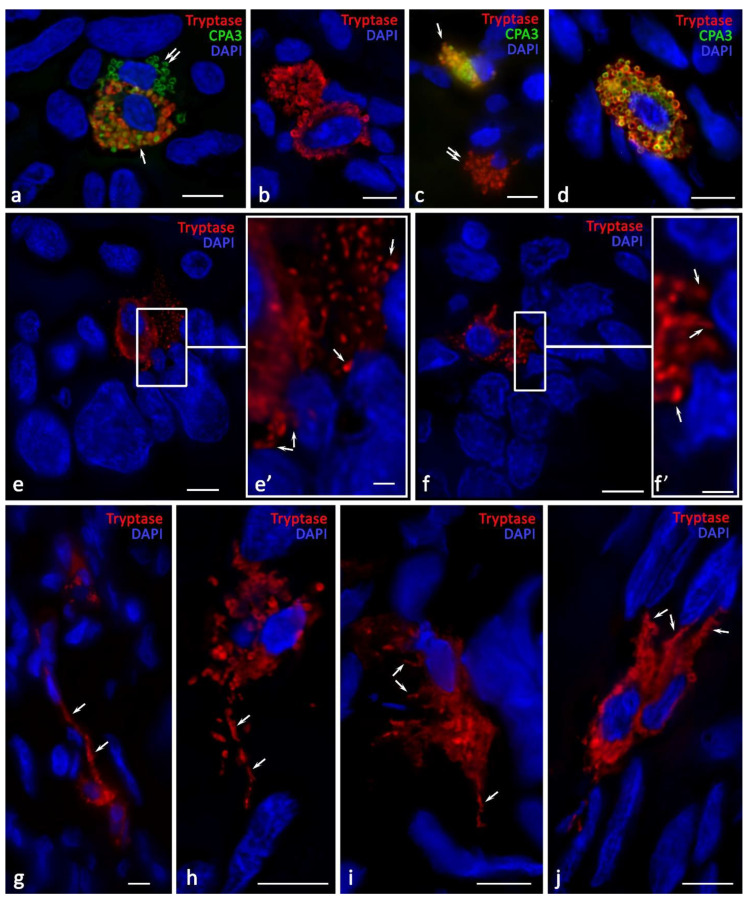
Histotopographic features of ovarian mast cells under endometrioid cyst development. (**a**–**c**) Colocalization of MCs with different phenotypes: Tr^+^CPA3^+^ (indicated by an arrow) and Tr^−^CPA3^+^ (indicated by a double arrow) (**a**), two MCs without CPA3 (**b**), Tr^+^CPA3^+^ (indicated by an arrow) and Tr^+^CPA3^−^ (indicated by a double arrow) (**c**). (**d**) A large MC with tryptase and CPA3 expression surrounded by several cells of the cytogenic stroma. (**e**,**f**) Options for MC contacting with the epithelium of endometrioid cysts. (**e′**) and (**f′**) are enlarged fragments of (**e**) and (**f**), respectively. The directed secretion of tryptase is clearly visible (indicated by an arrow), including that through the formation of thin MC outgrowths. (**g**–**j**) Morphological evidence of the formation of the MC cytoplasm outgrowths filled with tryptase and oriented towards other cells of the cytogenic stroma (indicated by an arrow). Scale: (**e′**,**f′**) 1 µm, the rest—5 µm.

**Figure 5 ijms-24-06498-f005:**
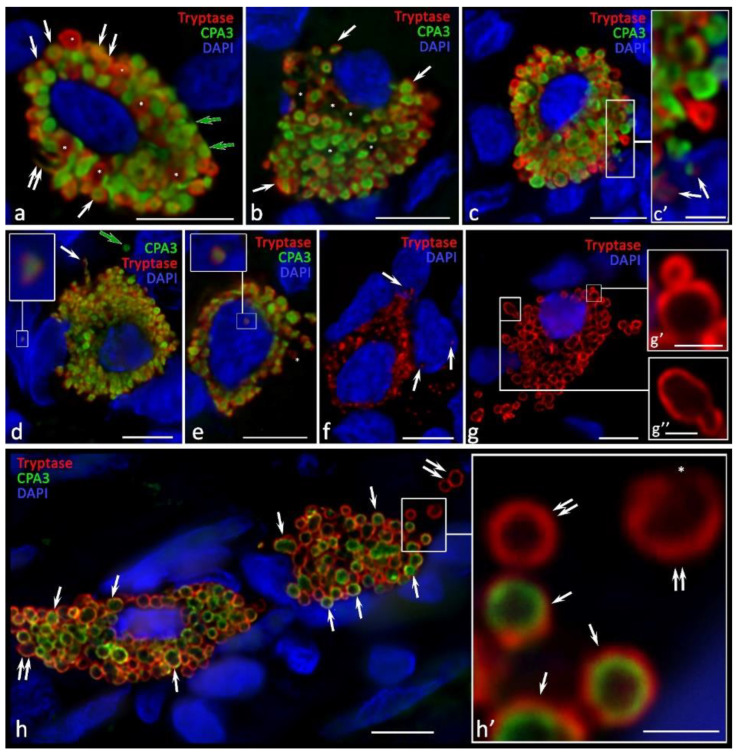
Cytotopography of specific proteases of ovarian mast cells with endometriomas. (**a**) MCs with a centrally located nucleus and large secretory granules with phenotypes Tr^+^CPA3^−^ (indicated by an asterisk), Tr^−^CPA3^+^ (indicated by a green arrow), and Tr^+^CPA3^+^ (indicated by an arrow). Thin cytoplasmic outgrowths immunopositive for tryptase and CPA3 are visible (indicated by a double arrow). (**b**) MCs with an eccentrically localized nucleus and a large number of secretory granules, with a predominant content of both specific proteases (indicated by an arrow). In the cytoplasm, there are areas that are not filled with specific proteases (indicated by an asterisk. (**c**) The secretion of tryptase and CPA3 is accompanied by transport to the nucleus of the neighboring cell (indicated by the arrow in (**c′**); (**c′**) is an enlarged fragment of (**c**)). (**d**) MCs with secretory granules of predominantly medium size, the cytoplasmic outgrowth is clearly visible in the direction of the neighboring cell (indicated by an arrow). A loose-lying granule in the intercellular matrix with the Tr^−^CPA3^+^ phenotype is visible (indicated by a green arrow). An enlarged fragment shows tryptase and CPA3 in the nucleus of a stromal cell. (**e**) MCs with a small number of secretory granules containing both specific proteases. An enlarged fragment shows tryptase and CPA3 as part of a granule in the MC nucleus. (**f**) MCs with small tryptase-positive formations in the cytoplasm, entering the extracellular matrix and penetrating the nuclei of other cells. (**g**) Tryptase-positive MCs with a clear localization of the protease in the peripheral region of the secretory granules. The mechanisms of secretory granules association (**g′**,**g″**) are clearly visible. (**h**) A large MC filled with granules (on the left) and a nuclear-free fragment filled with secretory granules, with a predominant Tr^+^CPA3^+^ phenotype (indicated by an arrow). Of a smaller number are granules that do not contain CPA3 (indicated by a double arrow). (**h′**) is an enlarged fragment of (**h**) showing the release of the secretome from the granule (indicated by an asterisk). Scale: (**c′**,**g′**,**g″**,**h′**) 1 µm, the rest—5 µm.

**Figure 6 ijms-24-06498-f006:**
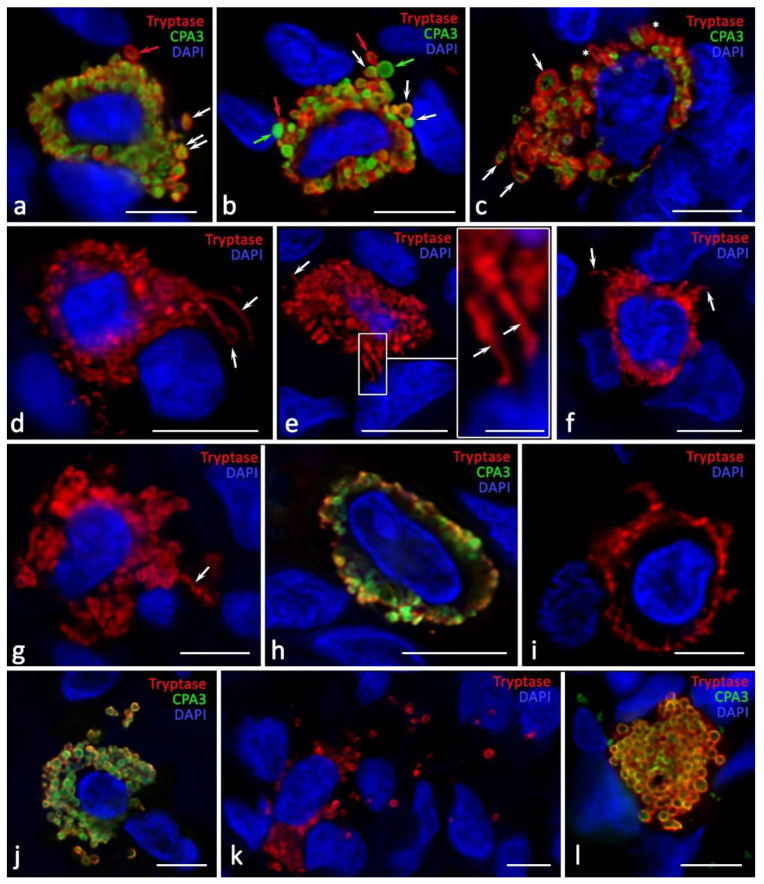
Secretory pathways of specific ovarian mast cell proteases in the development of endometriomas. (**a**–**c**) Entry of MC secretory granules of various phenotypes and sizes into the extracellular matrix: Tr^+^CPA3^−^ (indicated by a red arrow), Tr^−^CPA3^+^ (indicated by a green arrow), and Tr^+^CPA3^+^ (indicated by a white arrow). There is the potential of selective CPA3 secretion in the extracellular matrix from secretory granules with the Tr^+^CPA3^+^ phenotype ((**a**), indicated by a double arrow). Secretory granules can change their shape to be elongated in the direction of cellular targets ((**c**), indicated by an asterisk). (**d**–**g**) Formation of cytoplasmic outgrowths of various lengths and thicknesses (tunneling nanotubes), filled with proteases, to the targets of a specific tissue microenvironment (indicated by an arrow). (**h**,**i**) Morphological signs of secretion through the mechanism of transgranulation. (**j**,**k**) Entry of granules of various phenotypes into the extracellular matrix and their spread over large areas of the tissue. (**l**) Formation of large-sized nuclear-free fragments of the cytoplasm located in the local tissue microenvironment. Scale: 5 µm for the entire layout and 1 µm for the insert in (**e**).

**Table 1 ijms-24-06498-t001:** Number of mast cells in the ovary (per mm^2^).

Patient Groups	MC Histotopography	MC Specific Proteases	
		Tryptase-Positive MCs (M ± m)	Carboxypeptidase A3-Positive MCs (M ± m)
Patients with OEC (ovarian endometrioid cysts) (n = 27)	Medulla of the ovary	33.24 ± 4.57 *	41.21 ± 9.7 *
	The cortex of the ovary	3.38 ± 1.87 *	9.7 ± 2.1 *
	OEC wall	42.48 ± 5.67 *	47.5 ± 10.7 *
Control group (n = 6)	Medulla of the ovary	10 ± 1.72	8.0 ± 3.5
	Cortex of the ovary	2.75 ± 0.75	1.25 ± 0.9
	OEC wall	-	-

Notes: * *p* < 0.05 compared with the control group.

**Table 2 ijms-24-06498-t002:** Primary antibodies used in this study.

Antibodies	Host	Catalogue Nr.	Dilution	Sourse
Tryptase	Mouse monoclonal Ab	#ab2378	1:3000	AbCam, United Kingdom
Carboxypeptidase A3 (CPA3)	Rabbit polyclonal Ab	#ab251696	1:2000	AbCam, United Kingdom

**Table 3 ijms-24-06498-t003:** Secondary antibodies and other reagents.

Antibodies and Other Reagents	Source	Dilution	Label
Goat anti-mouse IgG Ab (#ab97035)	AbCam, United Kingdom	1/500	Cy3
Goat anti-rabbit IgG Ab (#ab150077):	AbCam, United Kingdom	1/500	Alexa Fluor 488
AmpliStain™ anti-Mouse 1-Step HRP (#AS-M1-HRP)	SDT GmbH, Baesweiler, Germany	ready-to-use	HRP
AmpliStain™ anti-Rabbit 1-Step HRP (#AS-R1-HRP)	SDT GmbH, Baesweiler, Germany	ready-to-use	HRP
4′,6-diamidino-2-phenylindole (DAPI, #D9542-5MG)	Sigma, Hamburg, Germany	5 µg/mL	w/o
VECTASHIELD^®^ Mounting Medium (#H-1000)	Vector Laboratories, Burlingame, CA, USA	ready-to-use	w/o
DAB Peroxidase Substrat Kit (#SK-4100)	Vector Laboratories, Burlingame, CA, USA	ready-to-use	DAB
Mayer’s hematoxylin (#MHS128)	Sigma-Aldrich	ready-to-use	w/o

## Data Availability

All data and materials are available on reasonable request. Address to (email: atyakshin-da@rudn.ru) or I.B. (email: buchwalow@pathologie-hh.de).

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
