# Peer review of "Mast Cell Tryptase and Carboxypeptidase A3 in the Formation of Ovarian Endometrioid Cysts"

_ijms, 2023, doi:10.3390/ijms24076498_

Round 1

Reviewer 1 Report

The manuscript is an original study, aligned to the IJMS Aims and Scope. Has a proper introduction, the results relies on histopathology evaluation, followed by an outstanding discussion. The paper is well written, has a smooth expression of ideas which makes it appealing. The major flaw is that authors gave no quantitative results. The image acquisition was performed with an advanced Zeiss system:  do the authors have quantitative measurements: fluorescence intensity/polarization, especially for the two target enzymes? The comparison between pathologic and control groups should be extended. As well, a table is needed, based on anonymized data, to provide a clear situation on the presence/absence of the two peptides in each of the 27 tumors (assessed by the three independent investigators). Alongside with some golden standard markers which were routinely measured for each patient, and some relevant clinical data for the subject addressed here. If feasible, a statistical analysis of these data, to enrich the Conclusions.  

 I suggest some minor changes as well, see below:  

Introduction, row 48:“ It has recently been demonstrated that the local tissue microenvironment of endometrioid lesions is characterized by features of cytokine and chemokine profiles that promote mast cell (MCs) recruitment with subsequent differenti-48 ation [14].”- please itemize the most important features

Materials and methods: 

Row 329: OEC abbreviation was not explained

Row 334: "OEC biomaterial from six patients without pathology was used as controls."- please explain how these samples were harvested, what was the medical suspicion to made the tissue resection in this cases? These control samples were previously archived?  

Row 358- Slide Stainer provider, city, country is missing

Row 358: “standard protocol”- reference 85 is a previous paper of the authors, not a golden standard for this procedure. A brief description of the method is needed, and authors should add to the reference list an independent source as well.

Row 387- please check Carl Zeiss Vision location: could be Aalen or Oberkochen in Germany instead Jena? ( Jena is headquarter of optical, not fluorescent devices and their companion software).

Figures 1-6: they are several high-quality images combined in the six cassettes, therefore the explanation of each part in the figure caption (a- j, a-r, a-v...) is very useful, but this generated a very long figure caption. Although an excellent description. Optionally, if some details could be moved to the main text of the Results section, without truncating the meaning, I recommend shorten the figure captions.

Author Response

Reviewer 1. The manuscript is an original study, aligned to the IJMS Aims and Scope. Has a proper introduction, the results relies on histopathology evaluation, followed by an outstanding discussion. The paper is well written, has a smooth expression of ideas which makes it appealing. The major flaw is that authors gave no quantitative results. The image acquisition was performed with an advanced Zeiss system:  do the authors have quantitative measurements: fluorescence intensity/polarization, especially for the two target enzymes? The comparison between pathologic and control groups should be extended. As well, a table is needed, based on anonymized data, to provide a clear situation on the presence/absence of the two peptides in each of the 27 tumors (assessed by the three independent investigators). Alongside with some golden standard markers which were routinely measured for each patient, and some relevant clinical data for the subject addressed here. If feasible, a statistical analysis of these data, to enrich the Conclusions. 

Authors:

Epifluorescence microscopy was not used in the study to quantify the expression of specific proteases in mast cells, but primarily to detect colocalization of tryptase and carboxypeptidase A3. To quantify the mast cell population in the ovary, we used a planimetric approach, determining the number of mast cells per mm2 of ovarian tissue. The morphometric analysis technique was performed using the open source software for digital pathology image analysis QuPath (Bankhead P., et al., 2017) and is given in the “Materials and Methods” section, in the section, and the results are presented in Table 1, in the “ Results".

Article citation: "Bankhead, P., Loughrey, M.B., Fernández, J.A. et al. QuPath: Open source software for digital pathology image analysis. Sci Rep 7, 16878 (2017). https://doi.org/10.1038/s41598-017-17204-5" has been added to the bibliography.

4.5. Image Acquisition

Stained tissue sections were observed on a ZEISS Axio Imager.Z2 equipped with a Zeiss alpha Plan-Apochromat objective 100×/1.46 Oil DIC M27, a Zeiss Objective Plan-Apochromat 150×/1.35 Glyc DIC Corr M27 and a ZEISS Axiocam 712 color digital microscope camera. Captured images were processed with the software program “Zen 3.0 Light Microscopy Software Package”, “ZEN Module Bundle Intellesis & Analysis for Light Microscopy”, “ZEN Module Z Stack Hardware” (Carl Zeiss Vision, Aalen, Germany) and submitted with the final revision of the manuscript at 300 DPI. Photomicrographs were obtained in some cases with a confocal microscope Nikon D-Eclipse C1 Si based on Nikon "Eclipse 90i". The number of mast cells in different areas of the ovary was determined using open source software for digital pathology image analysis QuPath (Bankhead P., et al., 2017) (See Table 1).

4.6. Statistical Analysis

Statistical analysis was performed using the SPSS software package (Version 13.0). The results are presented as mean (M) ± m (standard error of the mean). To assess the significance of the differences between the two groups, Mann–Whitney U test was used.

Reviewer. I suggest some minor changes as well, see below:  

1.2. Introduction, row 48:“ It has recently been demonstrated that the local tissue microenvironment of endometrioid lesions is characterized by features of cytokine and chemokine profiles that promote mast cell (MCs) recruitment with subsequent differentiation [14].”- please itemize the most important features

Authors:

добавлено в текст: “In particular, an increased level of SCF and CCL2 mRNA was formed, which was combined with an increase in the mRNA of the corresponding KITLG and CCR1 receptors, as well as an increase in the mRNA level of VCAM1, CPA3 and CMA1. At the same time, co-cultivation of mast cells with endometriotic epithelial cells and endometrial stromal cells led to increased production of pro-inflammatory and chemokinetic cytokines [14], which may explain the pathogenesis by the special properties of the integrative-buffer metabolic environment.

Materials and methods:

1.3.Row 329: OEC abbreviation was not explained

Authors: Corrected, OEC abbreviation added on line 18.

1.4.Row 334: "OEC biomaterial from six patients without pathology was used as controls."- please explain how these samples were harvested, what was the medical suspicion to made the tissue resection in this cases? These control samples were previously archived? 

Authors: Text changed: "The control group used biomaterial from 6 patients of childbearing age who died from diseases not associated with endometriosis or malignant neoplasms."

1.5.Row 358- Slide Stainer provider, city, country is missing

Authors: Automatic immunohistotainer Roche Ventana BenchMark Ultra, Roche Diagnostics, USA

1.6. Row 358: “standard protocol”- reference 85 is a previous paper of the authors, not a golden standard for this procedure. A brief description of the method is needed, and authors should add to the reference list an independent source as well.

Authors: Corrected: Included in the Materials and Methods section: Immunohistochemical visualization of bound primary antibodies was performed either with automatic immunohistotainer Roche Ventana BenchMark Ultra (Roche Diagnostics, USA) or manually, according to the standard protocol [83, 85]. For manually performed immunostaining, primary antibodies were incubated overnight at +4 °C at optimal dilution (Table 2).

.

1.7. Row 387- please check Carl Zeiss Vision location: could be Aalen or Oberkochen in Germany instead Jena? ( Jena is headquarter of optical, not fluorescent devices and their companion software).

Authors: The authors thank the referee for the remark. Indeed, Aaalen. Fixed: "Carl Zeiss Vision, Aalen, Germany"

Figures 1-6: they are several high-quality images combined in the six cassettes, therefore the explanation of each part in the figure caption (a- j, a-r, a-v...) is very useful, but this generated a very long figure caption. Although an excellent description. Optionally, if some details could be moved to the main text of the Results section, without truncating the meaning, I recommend shorten the figure captions.

Authors: The authors thank the referee for the high evaluation of the prepared microphotographs. In fact, when we selected the best and most informative photographs from the total array of acquired images, we found a large number of images containing valuable information about the biology of mast cells in a specific ovarian microenvironment. After receiving the reviewer's recommendations, we tried to reduce the legends as much as possible. The text for legends has been reduced, but, unfortunately, a significant reduction has not been achieved.

Reviewer 2 Report

Manuscript #ijms-2289078 explores the involvement of tryptase and Carboxypeptidase A3 of Mast Cell in “ovarian endometriosis”. The reported findings are in line with previous literature that reported the involvement of mast cells in formation of endometriosis (Biomedicine & Pharmacotherapy Volume 129, September 2020, 110476. https://doi.org/10.1016/j.biopha.2020.110476) and that activity of mast cells tryptase in involved in endometriosis and the associated angiogenesis (Experimental Cell Research Volume 332, Issue 2, 15 March 2015, Pages 157-162. https://doi.org/10.1016/j.yexcr.2014.11.014 and American Journal of Obstetrics and Gynecology Volume 193, Issue 6, December 2005, Pages 1961-1965. https://doi.org/10.1016/j.ajog.2005.04.055). The manuscript might be accepted after minor revision of the following issues:

1. Number and date should be given regarding the ethical approval from the Institutional Review Board.

2. For catalogue number of DAPI in table 2, is it necessary to indicate that you bought a 5 mg?

Author Response

Reviewer #2

Comments and Suggestions for Authors

Manuscript #ijms-2289078 explores the involvement of tryptase and Carboxypeptidase A3 of Mast Cell in “ovarian endometriosis”. The reported findings are in line with previous literature that reported the involvement of mast cells in formation of endometriosis (Biomedicine & Pharmacotherapy Volume 129, September 2020, 110476. https://doi.org/10.1016/j.biopha.2020.110476) and that activity of mast cells tryptase in involved in endometriosis and the associated angiogenesis (Experimental Cell Research Volume 332, Issue 2, 15 March 2015, Pages 157-162. https://doi.org/10.1016/j.yexcr.2014.11.014 and American Journal of Obstetrics and Gynecology Volume 193, Issue 6, December 2005, Pages 1961-1965. https://doi.org/10.1016/j.ajog.2005.04.055). The manuscript might be accepted after minor revision of the following issues:

The authors thank the reviewer for the analysis of the scientific manuscript. The answers are below.

2.1. Number and date should be given regarding the ethical approval from the Institutional Review Board.

Authors: Added to the Materials and Methods section: (approval protocol No. 3, 06/12/2019).

2.2. For catalogue number of DAPI in table 2, is it necessary to indicate that you bought a 5 mg?

Authors:  It can be helpful for the reader.

In addition, we have added important sources suggested by the reviewer to the list of references:

1.Li T, Wang J, Guo X, Yu Q, Ding S, Xu X, Peng Y, Zhu L, Zou G, Zhang X. Possible involvement of crosstalk between endometrial cells and mast cells in the development of endometriosis via CCL8/CCR1. Biomed Pharmacother. 2020 Sep;129:110476. doi: 10.1016/j.biopha.2020.110476. Epub 2020 Jul 8. PMID: 32768961.

  1. Ribatti D, Ranieri G. Tryptase, a novel angiogenic factor stored in mast cell granules. Exp Cell Res. 2015 Mar 15;332(2):157-62. doi: 10.1016/j.yexcr.2014.11.014. Epub 2014 Dec 3. PMID: 25478999.
  2. Ribatti D, Finato N, Crivellato E, Marzullo A, Mangieri D, Nico B, Vacca A, Beltrami CA. Neovascularization and mast cells with tryptase activity increase simultaneously with pathologic progression in human endometrial cancer. Am J Obstet Gynecol. 2005 Dec;193(6):1961-5. doi: 10.1016/j.ajog.2005.04.055. PMID: 16325597.

Reviewer 3 Report

Endometriosis is a chronic disorder characterized by the growth of endometrial-like tissue outside the uterine cavity and is found in approximately 10% of women of reproductive age. This study focuses on the mechanisms of endometrioid ovarian cyst formation or cystic ovarian endometriosis. Their founding indicates the potential role of mast cell tryptase and carboxypeptidase A3 in the development of ovarian endometriomas and infers new perspectives of their use as a pharmacological target in personalized medicine.The manuscript looks very well written, with clarity and a high degree of English, although small parts can be improved.In general, I do not have many comments, only the following.This is an excellent report dealing with significant technical matters. I find no fault whatsoever with the methods, data analysis, or conclusions.

Author Response

Reviewer #3

Comments and Suggestions for Authors

Endometriosis is a chronic disorder characterized by the growth of endometrial-like tissue outside the uterine cavity and is found in approximately 10% of women of reproductive age. This study focuses on the mechanisms of endometrioid ovarian cyst formation or cystic ovarian endometriosis. Their founding indicates the potential role of mast cell tryptase and carboxypeptidase A3 in the development of ovarian endometriomas and infers new perspectives of their use as a pharmacological target in personalized medicine.The manuscript looks very well written, with clarity and a high degree of English, although small parts can be improved.In general, I do not have many comments, only the following.This is an excellent report dealing with significant technical matters. I find no fault whatsoever with the methods, data analysis, or conclusions.

Authors: The authors thank the Reviewer for his support of the chosen direction of research and high appreciation of our study. We hope to continue this study in the future to more accurately detail the pathogenetic role of mast cells in the formation of endometrioid ovarian cysts.

Reviewer 4 Report

An interesting paper on the potential role of mast cell tryptase and carboxypeptidase A3 in the development of ovarian endometriomas, giving potential new perspectives of their use as a pharmacological target in personalized medicine.

I would suggest some minor revisions:

Introduction: lines 63-64. The meaning of the following sentence is not clear: "From these perspectives, the role of MCs in the pathogenesis of endometrioid cysts becomes obvious, including that due to the biological effects of specific MC proteases".

Line 144: plasma cells should be used instead of plasmocytes

line 156 "Apparently, these events are associated with the formation of fibrous....."( tissue should be added to the word fibrous).

Line 168 support should be corrected into supports

Abbreviations should be used along the manuscript; for instance MCs is used as the abbreviation for mast cells, but not along the whole manuscript

I would suggest to mention that the identification of mast cells during pathology examination of tissue specimens is often challenging and requires special stainings such as tryptase or CD117. For instance, in the gastrointestinal tract, mast cells infiltration (which may cause symptoms simulating various gastrointestinal diseases) can be extremely subtle (see reference Cancers 2021; 13(13):3316.)

Author Response

Reviewer #4  

Comments and Suggestions for Authors

Reviewer: An interesting paper on the potential role of mast cell tryptase and carboxypeptidase A3 in the development of ovarian endometriomas, giving potential new perspectives of their use as a pharmacological target in personalized medicine.

Authors:   The authors are grateful to the Reviewer for support of the translational significance of specific mast cell proteases and of the work performed.

Reviewer: I would suggest some minor revisions:

4.1. Introduction: lines 63-64. The meaning of the following sentence is not clear: "From these perspectives, the role of MCs in the pathogenesis of endometrioid cysts becomes obvious, including that due to the biological effects of specific MC proteases".

Authors:  In our work, we have sought to specifically note that mast cells have great potential to play key roles in the pathogenesis of endometrioid cysts, primarily due to the biological effects of specific proteases.

4.2. Line 144: plasma cells should be used instead of plasmocytes

Authors: Corrected.

4.3. line 156 "Apparently, these events are associated with the formation of fibrous....."( tissue should be added to the word fibrous).

Authors: Corrected.

4.4. Line 168 support should be corrected into supports

Authors: Corrected.

4.5. Abbreviations should be used along the manuscript; for instance MCs is used as the abbreviation for mast cells, but not along the whole manuscript

Authors: Corrected.

4.6. I would suggest to mention that the identification of mast cells during pathology examination of tissue specimens is often challenging and requires special stainings such as tryptase or CD117. For instance, in the gastrointestinal tract, mast cells infiltration (which may cause symptoms simulating various gastrointestinal diseases) can be extremely subtle (see reference Cancers 2021; 13(13):3316.)

Authors: Corrected, information added to the end of the discussion:

« It should be noted that when studying the role of mast cells in the ovary, the use of routine histochemical staining, for example, with toluidine blue, is not effective enough due to the low content of glycosaminoglycans in the secretory material for the development of the metachromasia effect. Therefore, for the identification of mast cells during pathology examination of tissue specimens, it is necessary to use characteristic markers that are found in most mast cells in their organ-specific population, including specific proteases or CD117. This conclusion is also objective for the pathologist to evaluate MCs in other or-gans, in particular, in the gastrointestinal tract [83].»

The list of references includes the article: «Zanelli M, Pizzi M, Sanguedolce F, Zizzo M, Palicelli A, Soriano A, Bisagni A, Martino G, Caprera C, Moretti M, Masia F, De Marco L, Froio E, Foroni M, Bernardelli G, Alvarez de Celis MI, Giunta A, Merli F, Ascani S. Gastrointestinal Manifestations in Systemic Mastocytosis: The Need of a Multidisciplinary Approach. Cancers (Basel). 2021 Jul 1;13(13):3316. doi: 10.3390/cancers13133316. PMID: 34282774; PMCID: PMC8269078»

Round 2

Reviewer 1 Report

I read the revised form of the manuscript and the author's response to my queries. Since all issues were addressed, I have no further comments.